# *Fusarium* Mycotoxins in Two Hulless Oat and Barley Cultivars Used for Food Purposes

**DOI:** 10.3390/foods9081037

**Published:** 2020-08-01

**Authors:** Ivana Polišenská, Ondřej Jirsa, Kateřina Vaculová, Markéta Pospíchalová, Simona Wawroszova, Jan Frydrych

**Affiliations:** 1Agrotest Fyto, Ltd., Havlíčkova 2787, 767 01 Kroměříž, Czech Republic; jirsa@vukrom.cz (O.J.); vaculova@vukrom.cz (K.V.); 2Central Institute for Supervising and Testing in Agriculture, Hroznová 2, 656 06 Brno, Czech Republic; marketa.pospichalova@ukzuz.cz (M.P.); simona.wawroszova@ukzuz.cz (S.W.); 3OSEVA Development and Research, Ltd., Hamerská 698, 756 54 Zubří, Czech Republic; frydrych@oseva.cz

**Keywords:** cereal safety, emerging mycotoxins, hulless barley, hulless oats

## Abstract

Hulless oats and hulless barley are highly valued for their excellent nutritional attributes and are increasingly being promoted in human nutrition. However, special attention should be paid to the risk of their contamination by *Fusarium* mycotoxins, as the rate of mycotoxin reduction during processing could be much lower than that for hulled cereals. In the present study, mycotoxin contamination of two cultivars, each of hulless oats and barley suitable for food purposes were studied in a 3-year field trial established in two contrasting environments. The contents of the mycotoxins regulated by law (deoxynivalenol and zearalenone) were low, and the present legal limits for their maximum content in unprocessed cereals were far from being exceeded. The mycotoxins most frequently occurring in hulless barley were enniatins (enniatin B, enniatin B_1_ and enniatin A_1_), beauvericin and nivalenol; hulless oats most frequently contained the HT-2 and T-2 toxins, beauvericin and enniatin B. The contents of enniatins and nivalenol were higher in barley than in oats. Close, positive relationships between the contents of the individual enniatins and between enniatins, beauvericin and nivalenol were observed, which implies that co-exposure could enhance the toxic potential of these mycotoxins through synergistic effects. The results highlight the need to pay more attention to the occurrence of enniatins, beauvericine and nivalenol in hulless oats and barley used for food purposes.

## 1. Introduction

Hulless oats and hulless barley are very convenient foodstuffs for enrichment of the carbohydrate group at the base of the generally recommended food pyramid as a more valuable substitute of the prevailing white wheat flour. Barley (*Hordeum vulgare* L.) is among the oldest domesticated plants. Its cultivation in the Fertile Crescent of the Middle East has been traced back approximately 10,000 years [1]. For thousands of years, barley was the dominant crop for feeding livestock, the production of fermented drinks, and also, especially in its hulless form (*Hordeum vulgare* L. var. *nudum* Hook. f.), use as human food. Oats (*Avena sativa* L.), the youngest of the grain species, were domesticated in Europe approximately 3000 years ago. Oats have been used mostly as animal feed but are also processed for human consumption. This kind of use, similar to that of barley, has often been connected with the hulless form, *Avena nuda* L., because this grain can be rolled or ground into flour with minimal processing, yielding a nutritious and flavorful foodstuff with a variety of uses.

Both hulless barley and hulless oats are considered to be important sources of numerous valuable substances of nutritional and biological importance, especially food fiber and β-glucan polysaccharides. According to Commission Regulation (EU) No 432/2012 [2], foods made from oats, oat bran, barley, barley bran, or mixtures of these sources containing at least 1 g of β-glucan per quantified portion contribute to the maintenance of normal blood cholesterol levels. A beneficial effect for consumers is achieved with a daily intake of 3 g of β-glucans.

As is true for all other kind of cereals, it is important to take into consideration the potential for contamination of hulless barley and oats with *Fusarium* mycotoxins. These toxins are produced by many *Fusarium* species and cause the plant disease *Fusarium* head blight (FHB). There are three main negative consequences of FHB infection in cereals: loss of grain yield, impaired technological quality, and contamination by *Fusarium* mycotoxins. The composition of the *Fusarium* pathogen complex occurring on cereal heads is variable and influenced by many factors, such as weather conditions, location, cropping practices, and cereal species, as different cereal types can bear different *Fusarium* species spectra [3]. Most studies concerning FHB have focused on wheat, but *Fusarium* spp. pathogens commonly infect barley [4,5,6], oats [3,4,7], and other cereal species. As a consequence of the variability in the occurrence of *Fusarium* pathogens, there also exists a broad variability in mycotoxin spectra and concentrations.

Modern analytical techniques can determine a multitude of fungal metabolites that are biosynthesized by *Fusarium* spp. associated with FHB infection in cereals. In addition to known mycotoxins, for which maximum levels in food are enforced, currently unregulated, so-called emerging mycotoxins have been shown to occur frequently in agricultural products [8]. The European Union has established maximum limits (MLs) for the *Fusarium* mycotoxins deoxynivalenol (DON) and zearalenone (ZEA) in foodstuffs, including cereals and cereal products (Commission Regulation (EC) No. 1881/2006) [9]. There is a different ML for DON content in unprocessed durum wheat, oats, and maize (1750 μg kg^−1^) from that in the other cereals, including common wheat and barley, which is 1250 μg kg^−1^. The ML for ZEA is equal for all kinds of unprocessed cereals (100 μg kg^−1^), with the exception of maize (350 μg kg^−1^). Several years ago, the EU recommended indicative levels for the sum of T-2 toxin (T2) and HT2 toxin (HT2), above which, mainly in the case of repetitive findings, investigations should be carried out to determine the factors that lead to the presence of these mycotoxins. The indicative levels are not feed and food safety levels [10]. For unprocessed barley, the indicative level for the sum of T2 and HT2 was set to 200 μg kg^−1^, and for oats, this level was set to 1000 μg kg^−1^. Apart from DON, ZEA, T2 and HT2, the tolerable daily intake was also established for fumonisins and nivalenol (NIV) [9]; however, until now, no specific limit was set for NIV and for fumonisins—it was set for maize only. Among emerging mycotoxins, *Fusarium* metabolites such as fusaproliferin, beauvericin (BEA), enniatins (ENs), and moniliformin are those that are mentioned most often [8,11]. There are limited data on the toxicity, occurrence, and contamination levels of these metabolites. The European Food Safety Authority (EFSA) highly recommends monitoring ENs and BEA in food and feed by means of liquid chromatography with tandem mass spectrometry (LC-MS/MS) methods and studying their co-occurrence with other *Fusarium* toxins because of possible combined effects [12].

In the case of hulled barley and oats, grain is subjected to a dehulling process before it is used for food production. This can substantially decrease mycotoxin content. According to Scudamore et al. [13], the concentrations of each *Fusarium* mycotoxin studied (HT2 and T2, ZEA and DON) decreased by 90–95% during oat processing, particularly after the dehulling step, because the toxins are mostly concentrated in the hulls. In the case of hulless cereals, no such process is applied to the grain with the exception of cleaning and, in some cases, scouring. Therefore, special attention should be given to hulless cereals, as the extent of reduction during their processing is much less than that for hulled cereals. The maximum EU limits for mycotoxins do not distinguish between the unprocessed hulled cereals versus hulless cereals and are applied to unprocessed cereals ‘such as’, i.e., to the cereals before the first-stage processing.

Information about the relative susceptibilities of hulless versus hulled cultivars of barley and oats to mycotoxin contamination is scarce and diverse, and mainly focus on DON. Berger et al. [14] and He et al. [15] did not find a significant difference in FHB incidence and DON accumulation between hulled and hulless barley genotypes. Legzdina and Buerstmayr [16] reported that the DON accumulation of hulled barley was significantly higher than that of hulless barley, whereas for NIV, there was no significant difference between the mean values of covered and hulless barley. Malachová et al. [17] reported higher levels of mycotoxins (DON, T2, HT2, NIV) in hulless cultivars. For oats, it has been reported that hulless oat genotypes accumulate lower amounts of DON than hulled oats [18,19,20,21]. After dehulling, the grain of hulled oat cultivars contained less DON in their dehulled kernels than the grain of hulless oat cultivars [19], which was similarly observed for barley [4]. Information about contamination of hulless barley and oats with respect to some emerging mycotoxins is entirely missing.

The aim of our study was to investigate mycotoxin contamination of hulless barley and hulless oats grown under different environments and to compare the concentrations with the legal limits for cereals intended for food use (if existing).

## 2. Materials and Methods

### 2.1. Plant Material and Field Trials

Small-Parcel experiments (10 m^2^, four randomized replications) were conducted in fields of the Research Institute in Kroměříž (KM) and the Research Institute in Zubří (ZB), both in the Czech Republic, during harvest years 2015, 2016, and 2017. Kroměříž is located within a sugar beet agricultural production area in a warm and moderately wet region, while Zubří is situated in a slightly warm climatic region. Characteristics of these locations are shown in Table 1, together with their climatic conditions during the growing season and the dates of sowing and harvest. In all experimental years, experimental site KM had higher mean temperature and lower sum of rainfall compared with experimental site ZB. The sum of rainfall was the highest in 2016 and lowest in 2015 at both experimental sites. Temperature was the highest in 2017 at both experimental sites; KM had comparable temperatures in 2015 and 2016, and ZB had a lower temperature in 2015 than 2016. The trial included two hulless spring barley cultivars (in parentheses—maintainer, registration year): AF Cesar (Agrotest Fyto, Ltd., Kroměříž, Czech Republic, 2014) and AF Lucius (Agrotest Fyto, Ltd., Kroměříž, Czech Republic, 2009) and two hulless spring oat cultivars: Otakar (Selgen, Plc., Prague, Czech Republic, 2011) and Saul (Selgen, Plc., Prague, Czech Republic, 2005). All are convenient and used for food purposes. Sowing was carried out using an Oyjord-type sowing machine (Wintersteiger, Ried im Innkreis, Austria). In all three years at both locations, rapeseed had been the preceding crop. Meteorological parameters were monitored by meteorological stations approximately 500 m from the field experiments at KM and ZB. Treatments with herbicides, insecticides, and growth regulators were performed according to the situation specific to the given location and time while following good principles of rational plant protection. All plots were treated with fungicides against leaf diseases at the beginning of stem elongation. The total dose of nitrogen was 60 kg ha^−1^ (20 kg ha^−1^ preplant, 40 kg ha^−1^ early topdress). The experimental plots were harvested using an Osevan S 03-060 small-plot harvester (Oseva, Litomyšl, Czech Republic) at full maturity with each replication harvested individually. The harvested grain was cleaned using a Petkus K 541 cleaner/sorter (PETKUS Technologie GmbH, Wutha-Farnroda, Germany) with a 1-mm screen, carefully mixed, and subsamples for milling were taken using a sample divider.

### 2.2. Preparation of Samples and Analysis of Mycotoxins by UPLC/MS/MS

Mycotoxins were analyzed in whole meal flour obtained using a sample mill (Pulverisette 19, Fritsch, Idar-Oberstein, Germany) with a 1-mm screen, each field replication separately. Samples were stored at −20 °C until analysis. Samples were prepared using the modified QuEChERS method [22]. Before weighing (2 g) samples for extraction, each sample was carefully mixed to ensure homogeneity. The homogenous sample was extracted with acetonitrile and water. The mixture was shaken intensively for 20 min, and after the addition of NaCl and MgSO_4_, the mixture was shaken by hand for 1 min and then centrifuged (5 min, 5000 rpm). An aliquot of the organic phase was cleaned by freezing out for at least 2 h and then centrifuged (5 min, 5000 rpm). A 0.5 mL aliquot of the organic phase was diluted with 0.5 mL of deionized water, mixed and filtered through a 0.2-µm nylon membrane filter. Mycotoxin analysis was performed using a Waters Acquity UPLC system coupled to a Xevo TQ MS (Waters, Milford, CT, USA) triple quadrupole mass spectrometer equipped with an electrospray ion source operated in positive mode. The acquisition of data was performed in multiple reaction monitoring mode. Two product ions (one quantifier, one qualifier) were monitored for each mycotoxin. Details about chromatographic separation, mass spectrometric conditions, and validation parameters of multiple mycotoxin determination in cereals were described in a previous study [22]. Samples were analyzed in duplicates.

The following mycotoxins were analyzed (in parentheses—limit of quantification): DON (50 μg kg^−1^), ZEA (20 μg kg^−1^), NIV (80 μg kg^−1^), T2 (5 μg kg^−1^), HT2 (5 μg kg^−1^), BEA (5 μg kg^−1^), enniatin A (ENA; 5 μg kg^−1^), enniatin A_1_ (ENA1; 5 μg kg^−1^), enniatin B (ENB; 5 μg kg^−1^), enniatin B_1_ (ENB1; 5 μg kg^−1^), fumonisin B_1_ (FB1; 10 μg kg^−1^), and fumonisin B_2_ (FB2; 20 μg kg^−1^). The frequency of mycotoxin occurrence is expressed as the percentage of samples with a given mycotoxin above the limit of quantification (LOQ).

### 2.3. Data Analyses

Analytical data are reported as the mean ± standard deviation of four field replications. Results reported as below their LOQ were replaced by half their respective LOQ (middle-bound estimate). Normality of data was assessed using the Shapiro–Wilk test. Due to non-normality of both natural and log-transformed data sets in some cases, non-parametric methods were used. Statistical comparison of the means was made by the Friedman test (years), the Wilcoxon signed-rank test (localities, cultivars) or the Mann–Whitney U Test (crops). Correlations between the parameters were determined using the Spearman’s correlation test. All calculations were performed using the software package *Statistica*, version 12 (StatSoft Inc., St Tulsa, OK, USA). The significance level was set at *p* < 0.05.

## 3. Results

Mycotoxins ZEA and FB2 were not detected in any grain sample. FB1 occurred only in one oat and one barley sample. Therefore, ZEA, FB1 and FB2 are not listed in the resulting graphs and tables and were not included in the statistical calculations. The presence of FB1 is commented on in the appropriate paragraph.

### 3.1. Content of Mycotoxins in Barley

In barley grain, the most frequently occurring mycotoxin was ENB, which was present in all 12 barley samples (three harvest years × two cultivars × two locations), followed by enniatin ENB1 (present in 92% of barley samples), ENA1 (75%) and BEA (75%) (Appendix A). NIV was found in 58% of the samples and HT2 in 50%. The highest concentrations were found for NIV (mean 239 ± 218 μg kg^−1^; max 728 μg kg^−1^) and ENB (mean 226 ± 239 μg kg^−1^; max 592 μg kg^−1^). The concentrations of the other mycotoxins were much lower (in parentheses—mean; maximum): ENB1 (105 ± 105 μg kg^−1^; 281 μg kg^−1^), BEA (57 ± 118 μg kg^−1^; 423 μg kg^−1^), ENA1 (32 ± 32 μg kg^−1^; 87 μg kg^−1^), HT2 (10 ± 9 μg kg^−1^; 25 μg kg^−1^) and ENA (5 ± 4 μg kg^−1^; 13 μg kg^−1^). DON was present in 2 out of 12 (17%) barley samples, reaching values of 50 μg kg^−1^ and 66 μg kg^−1^, whereas ZEA was not detected at all, similar to T2 and FB2. FB1 was found in one barley sample (327 μg kg^−1^) at experimental place ZB from the 2016 harvest.

The most influential factor was shown to be the location of growth (Table 2). The mean concentrations for ENB, ENB1, NIV, BEA and ENA1 were significantly higher at ZB, and at KM for HT2 (Figure 1a). The harvest year significantly influenced the ENB, BEA, ENB1 and ENA1 content, which all were higher in 2016. There was no difference between the two cultivars for any of the detected mycotoxins (*p* > 0.180).

### 3.2. Content of Mycotoxins in Oats

In oats, the most frequently occurring mycotoxin was HT2, which was present in 7 out of 12 oat samples (three harvest years × two cultivars × two locations; 58% of oat samples) (Appendix A). T2, BEA and ENB were found in 33% of oat samples, and NIV and ENB1 were found in 17% of oat samples. The highest maximum value was found for NIV (304 μg kg^−1^), but it was detected in only two oat samples; therefore, the mean value was below the LOQ. The concentrations of other mycotoxins were quite low: (in parentheses—mean; maximum) HT2 (11 ± 9 μg kg^−1^; 28 μg kg^−1^), BEA (10 ± 12 μg kg^−1^; 35 μg kg^−1^), ENB (10 ± 17 μg kg^−1^; 55 μg kg^−1^), T2 (<LOQ; 11 μg kg^−1^), ENB1 (<LOQ; 15 μg kg^−1^). The legislatively limited DON and ZEA were not present in any oat sample, similar to ENA, ENA1 and FB2. FB1 was found in a single oat sample at the level of 19 μg kg^−1^ at experimental place ZB from the 2016 harvest. The location of growth significantly influenced only the contents of HT2 (Table 2), which was significantly higher at KM (Figure 1b). Harvest year did not influence the content of mycotoxins in oats. There was also no difference between cultivars for any of the detected mycotoxins (*p* > 0.180).

### 3.3. Comparison of Mycotoxin Occurrence in Barley and Oats

Considering the data set as a whole (three harvest years × two cultivars of each crop × two experimental sites), significantly higher contents of ENB, ENB1, ENA1, and NIV were found in barley compared with oats (Table 3). Comparing the contamination of oats and barley at individual experimental sites, at ZB, significantly higher contents of ENB, ENB1, NIV, BEA and ENA1 were found in barley compared to oats. At KM, higher contents of ENB and ENB1 in barley were observed.

### 3.4. Relationship between Individual Mycotoxins

Significant positive correlations between NIV, BEA, and all ENs were found (Table 4). Apart from the relationships between some of the ENs (ENB and ENB1; ENA1 and ENB1; ENA1 and ENB), the strongest positive correlation was observed between NIV and BEA (*r* = 0.833). Significant negative relationships were observed between HT2 and NIV (*r* = −0.517) and between HT2 and BEA (*r* = −0.495).

## 4. Discussion

In the current study, mycotoxin content in the two cultivars of hulless oats and hulless barley was studied in a 3-year field trial established at two experimental sites characterized by contrasting environmental conditions. Apart from the legislatively regulated contents of DON and ZEA, mycotoxins recurrently discussed as eligible for regulation (T2, HT2, NIV, fumonisins), and emerging mycotoxins BEA and ENs, were analyzed in the harvested grain.

### 4.1. Deoxynivalenol (DON) and Zearalenone (ZEA)

Evaluating data of the two experimental sites together, DON occurred in two of 12 hulless barley treatments (three harvest years × two cultivars × two locations) (17%) at a maximum concentration of 66 μg kg^−1^. In hulless oats, DON was not found in any harvest year, cultivar, or location. In general, for commonly grown hulled type of barley and oats, lower concentrations of DON in oats compared to barley have mostly been reported [3,23,24,25]. For example, Edwards [24] found that among barley and oats harvested in UK between 2002–2005, 57% of barley and 32% of oat samples contained DON, with maximum level in barley of 1416 and 282 μg kg^−1^ in oats. Similarly, Schöneberg et al. [3,25] found a higher maximum DON level in barley (4860 μg kg^−1^) than in oats (1328 μg kg^−1^), with the frequency of occurrence higher in barley (57%) [25] than in oats (45%) [3].

The second regulated mycotoxin, ZEA, was not found in any of the hulless barley or oat samples. ZEA is often reported to be low both in hulled barley [24,25,26] and oats [3,27]. For example, among 296 oat samples harvested in the UK, only 1% of the samples had a ZEA concentration greater than 10 μg kg^−1^, and both the mean and median were below the LOQ (3 μg kg^−1^). Similarly, among 339 barley samples, only 2% were greater than 10 μg kg^−1^, and both the mean and median were below the LOQ (3 μg kg^−1^) [26].

To summarize the results for both of the legislatively regulated mycotoxins DON and ZEA in hulless barley and oats, the present legal limits [9] for maximum mycotoxin content in unprocessed cereals for food purposes (DON in barley of 1250 μg kg^−1^, DON in oats of 1750 μg kg^−1^, ZEA in both barley and oats of 100 μg kg^−1^) were far from being exceeded.

### 4.2. Fumonisins and T-2 and HT-2 Toxins

Fumonisins are legislatively limited as the sum of FB1 and FB2, in food maize [9] only, with a maximum limit of 4000 μg kg^−1^. In our study, FB1 was found in one barley sample (327 μg kg^−1^) and one oat sample (19 μg kg^−1^), both of which were harvested at experimental site ZB in 2016. Fumonisins are currently found mainly in maize because maize is the preferred host of pathogens producing these mycotoxins, such as *F. proliferatum* and *F. verticillioides*. In barley, fumonisins were found by Beccari et al. [28] in 2% of samples, with concentrations at the levels of 156 μg kg^−1^ for FB1 and 65 μg kg^−1^ for FB2. In oats, fumonisins have only seldom been reported [29]. Our results are in a good agreement with these findings and confirmed that fumonisins do not pose a substantial risk for hulless barley and oats.

The maximum indicative value for the sum of T2 and HT2 is different for barley (200 μg kg^−1^) and for oats (1000 μg kg^−1^) [10]. In our study, T2 was not found in any of the barley samples; therefore, the maximum HT2 concentration found (25 μg kg^−1^) constituted the maximum combined T2 and HT2 concentration. In oats, both T2 and HT2 were found, with a maximum sum of T2 and HT2 of 30 μg kg^−1^. Thus, in both barley and oats, the maximum values reached were much lower than the maximum indicative values. Our results agree with other findings that T2 occurred less often and at lower concentrations compared with HT2 [28,30,31]. Toxins T2 and HT2 are often reported to be mycotoxins with the highest occurrence and concentration in oats [3,31,32,33]. In our trial, this was true at the KM site, where T2 and HT2 were the only mycotoxins that were found in oats. At experimental site ZB, BEA and ENB were found more often than T2 and HT2, and their concentrations were also higher. Similarly, Fredlund et al. [32] found ENs and BEA more often and in higher concentrations in oats harvested in Sweden than HT2, DON, and T2.

### 4.3. Nivalenol (NIV)

The maximum concentration of all mycotoxins analysis was found for NIV in both barley (maximum value 727 μg kg^−1^) and oats (maximum value 304 μg kg^−1^). Evaluating data from the two experimental sites together, NIV was found in 58% of the barley samples and in 17% of the oat samples, and the contamination level was significantly higher in barley than in oats. NIV is commonly found in barley [28,30,34]. For example, Beccari et al. [28] found NIV in 35% of barley samples with a maximum concentration of 434 μg kg^−1^, and Nielsen et al. [34] reported a maximum concentration of 1089 μg kg^−1^. The occurrence of NIV in oats is also often reported and even ranked second [3,31,33] or third [21] in occurrence after T2 and HT2. Both Edwards [24] and Schöneberg et al. [3,25] found higher concentrations of NIV in oats than in barley, with contamination levels greatly affected by the harvest year. Based on our results, NIV content was influenced by the location of growing. NIV, when orally ingested by animals, is more toxic than DON [35]. The European Food Safety Administration established a lower tolerated daily intake of 0.7 μg kg^−1^ body weight for NIV compared with 1 μg kg^−1^ for DON [9]. There is, however, no legislatively defined limit for NIV in food cereals.

### 4.4. Enniatins (ENs) and Beauvericine (BEA)

In hulless barley harvested in our trial, of all the mycotoxins analyzed, the most abundant mycotoxin was ENB, which was detected in all samples grown at both experimental sites. Similarly, in a survey of cereals harvested from common farm fields in Denmark, Svingen et al. [36] detected ENB in all 110 tested cereal samples, including 56 barley and 11 oat samples. They reported the level of contamination with the individual ENs in the order of ENB > ENB1 > ENA1 > ENA, which fully agrees with our results. Similar contamination order levels in barley by the individual ENs was also reported by Beccari et al. [28]. They determined a maximum concentration of 171 μg kg^−1^ for ENB and 101 μg kg^−1^ for ENB1 in barley harvested in Italy, which is less than the results we found in barley grown at ZB (ENB, 592 μg kg^−1^ and ENB1, 281 μg kg^−1^) but more than in barley grown at KM (ENB, 86 μg kg^−1^ and ENB1, 70 μg kg^−1^). In Danish barley, Svingen et al. [36] determined maximum concentrations of ENB up to 2100 μg kg^−1^ and of ENB1 up to 520 μg kg^−1^. In oats, the concentrations of ENs were lower than those in barley, reaching maximum concentrations of 55 μg kg^−1^ for ENB and 15 μg kg^−1^ for ENB1, and were found at a lower frequency; ENB was detected in 33% and ENB1 in 17% of oat samples. Lower concentrations of ENs in oats compared with barley were also found by Svingen et al. [36] and Bryla et al. [37].

BEA was present in 75% and 33% of barley and oat samples, respectively, and the concentrations in barley were higher (maximum of 423 μg kg^−1^) than those in oats (maximum of 35 μg kg^−1^). In contrast, Svingen et al. [36] found BEA more often in oats (in 73% of oat samples) than in barley (in 7% of samples), with maximum concentrations that were similar (max of 130 μg kg^−1^ in barley and max 110 μg kg^−1^ in oats). The BEA concentrations found in barley in the current study are higher than those found more recently in Denmark [36] and Italy [28] (max 316 μg kg^−1^) and in the studies reviewed in the EFSA report [12] (max 69 μg kg^−1^). BEA was formerly reported as an important cereal contaminant mainly from Finland and other Nordic countries [38,39,40]. On the other hand, some of the authors reported a higher contamination level in Southern Europe and Morocco [41]. We found significantly higher BEA contamination in both barley and oats at the colder and wetter experimental site ZB. Our results for both barley and oats correspond with those of Covarelli et al. [42], that ENs dominate over BEA, and even if their frequencies are similar, the ENs contamination levels are higher than those of BEA.

ENs have recently been shown to be one of the most prevalent emerging mycotoxins across geographical regions [36]. ENs are cytotoxic and have antibacterial, anthelmintic, antifungal, herbicidal, and insecticidal effects [43]. In an in vitro quadroprobe assay, enniatin B was more toxic than aflatoxin B1 [36]. Although ENs have been proven to be toxic in vitro, most in vivo data indicate no or only low toxicity [8]; therefore, ENs are currently considered mainly for their combined exposure, which can reach levels that are of concern for chronic exposure to humans and animals [12]. However, given the lack of relevant toxicity data, no firm conclusion could be drawn, and research into their toxicological effects is still ongoing [43]. The chemical structure of ENs is similar to that of BEA, both of which are cyclic hexadepsipeptides [44]. A high co-occurrence of ENA, ENA1, ENB and ENB1 and co-occurrence of BEA and ENs have been confirmed in some previous studies and were observed also in our trial. The co-occurrence is explained by the fact, that these mycotoxins are structurally related and produced by the same *Fusarium* species through the same metabolic pathway [11,45]. The co-occurrence of BEA and ENs with other *Fusarium* toxins, such as DON, moniliformin and fumonisins, has also been reported [12]. We observed, apart from above mentioned relationship between the individual ENs, and ENs and BEA, also a positive correlation between ENs and NIV, and NIV and BEA but not between ENs, DON and fumonisins. On the other hand, we observed a significant negative relationship between NIV and HT2, and BEA and HT2. This might imply that the producer/producers of ENs, BEA and NIV are different from those of HT2.

### 4.5. The Main Differences between Hulless Barley and Oats

To summarize the differences between hulless oats and hulless barley, the frequency of occurrence of the individual mycotoxins was as follows in hulless barley: ENB > ENB1 > ENA1 > BEA > NIV > HT2 > ENA > DON > T2; in hulless oats, this order was HT2 > T2 = BEA = ENB > ENB1 = NIV > DON = ENA = ENA1. Significantly higher mean concentrations of ENB, ENB1, ENA1 and NIV were found in hulless barley than in hulless oats. Although it is known that different small-grain cereal species could bear a different *Fusarium* species spectrum, high seasonal and regional variability should be taken into consideration. As was shown by Langseth and Elen [46], weather conditions and different local agrotechnical measures used for oats and barley can influence mycotoxin content to a greater extent than differences between these crops itself.

### 4.6. Factors Influencing Mycotoxin Content of Hulless Barley and Oats

The location of growth was the most influential factor of mycotoxin contamination for both barley and oats. For barley, at the experimental site ZB, ENs, BEA, and NIV were found more frequently and at higher concentrations than those found at KM. On the other hand, at KM, the contamination level of HT2 was higher, which was proved for both barley and oats. The experimental site ZB is characterized by a higher altitude, harsher weather and poorer soil conditions compared with KM, and during all experimental years, a higher sum of rainfall and a lower temperature in the vegetation seasons were recorded at ZB. Harvest year significantly influenced the contents of ENs in barley, being higher in 2016, which had the highest sum of rainfall during the vegetation season. The association of a higher occurrence of ENs with the harvest year characterized by the highest sum of rainfall corresponds with the fact that ENs were more abundant at the colder and wetter experimental site ZB. It confirms that the geographical factors, including climate, are of superior importance for the occurrence of FHB and for the pattern of infestation by various *Fusarium* species [46]. The main ENs producers are *F. avenaceum* [47,48], which can also produce moniliformin and BEA [49], *F. poae*, which produces NIV, BEA, and fusarin in addition to ENs [50,51], and also *F. tricinctum* [8]. Both *F. avenaceum* and *F. poae* have been reported to be better adapted to cooler conditions [52]. Nevertheless, as stated by Uhlig et al. [40], *F. avenaceum* can be isolated from grain over a range of climatic zones, and both *F. avenaceum* and *F. poae* were recently found to be the predominant species on malting barley in central Italy [28]. This could also explain the high content of BEA found in Southern Europe and Morocco [41]. Covarelli et al. [42] reported that *F. poae* and *F. avenaceum* increased their presence when climatic conditions were not favorable for the development of the main FHB causal agents, such as *F. graminearum*, the main DON and ZEA producer. As suggested by Nielsen et al. [34], *F. graminearum*, and particularly *F. culmorum,* are not the most important pathogens as part of the FHB complex in barley in Europe, and research focus should be directed towards understanding the impact of other species previously considered to be less aggressive. This is in agreement with our results, as we found the mycotoxins DON and ZEA, produced by *F. graminearum* and *F. culmorum* less often and in lower concentrations compared with those produced by *F. poae* and *F. avenaceum*, such as ENs, BEA and NIV, in both hulless barley and hulless oats.

## 5. Conclusions

To our knowledge, this is the first study to address emerging mycotoxins in hulless oats and barley. High levels of ENs, BEA and NIV were found in hulless barley, and their occurrence was promoted by an environment characterized by higher rainfall and lower temperature during the vegetation period. Although these mycotoxins were also detected in hulless oats, their contents were lower than those in hulless barley. The presence of ENs, BEA and NIV were mutually positively correlated, which can imply potential for the combined risk leading to simultaneous toxicological effects after consumption. As these mycotoxins are not currently regulated, they are not regularly monitored in cereals intended for food production. This may be even more important for hulless cereals because they are not dehulled before processing and, therefore, more mycotoxins can be transferred from the raw cereals into the final product. The contents of DON and ZEA, which are currently limited by legislation, were low in hulless barley and even lower in hulless oats. These results highlight the need to pay more attention to the occurrence of ENs, BEA and NIV in hulless oats and hulless barley used for food purposes.

## Figures and Tables

**Figure 1 foods-09-01037-f001:**
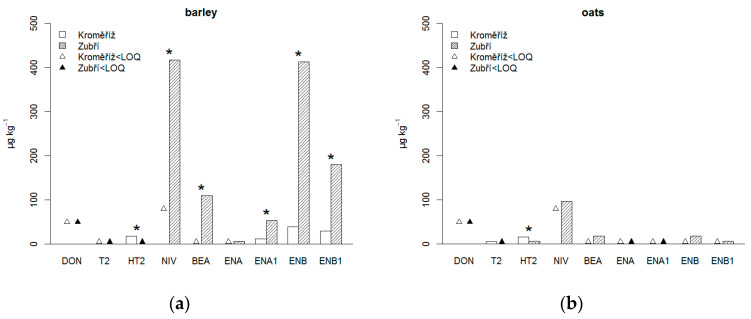
Mycotoxin content in hulless barley (**a**) and oats (**b**) grown at two experimental sites (Kroměříž, Zubří) (mean of two cultivars of each crop grown in three harvest years). Significant differences between locations are marked as (*) for *p* < 0.05. DON–deoxynivalenol, T2–T2-toxin, HT2–HT2-toxin, NIV–nivalenol, BEA–beauvericin, ENA–enniatin A, ENA1–enniatin A_1_, ENB–enniatin B, ENB1–enniatin B_1_.

**Table 1 foods-09-01037-t001:** Characteristics of experimental sites and sowing and harvest dates of field trials during harvest years 2015–2017.

Locality		Kroměříž			Zubří	
Latitude, longitude	49°17′ N, 17°22′ E			49°28′ N, 18°5′ E		
Altitude (m a.s.l.)		235			345	
Average annual temperature ^1^		9.2 °C			7.5 °C	
Average total annual precipitation ^1^		576 mm			865 mm	
Soil type	Luvic Chernozem			Gleyic Fluvisol		
Soil textural class	Silty clay loam			Sandy loam		
Year	2015	2016	2017	2015	2016	2017
Mean temperature ^2^	13.9 °C	13.9 °C	14.4 °C	13.0 °C	13.2 °C	13.4 °C
Rainfall ^2^	184 mm	309 mm	248 mm	213 mm	468 mm	383 mm
Date of sowing	24-March	30-March	28-March	14-Apr	5-Apr	30-March
Date of harvest-barley	30-July	27-July	22-July	3-August	9-August	24-July
Date of harvest-oats	5-August	30-July	23-July	3-August	9-August	1-August

^1^ Average based on period 1971–2010; ^2^ mean daily temperature and sum of precipitation from 21 March to 10 August.

**Table 2 foods-09-01037-t002:** The influence (*p* values) of harvest year, location and cultivar on mycotoxin content in hulless barley and oats.

			*p* Values for Concentrations								
Crop	Factor	*n*	DON	T2	HT2	NIV	BEA	ENA	ENA1	ENB	ENB1
barley	Year ^a^	3	0.607	0.368	0.607	0.761	***0.032***	0.368	***0.050***	***0.018***	***0.039***
	Location ^b^	1	0.180	n.a.	***0.028***	***0.028***	***0.028***	0.109	***0.043***	***0.028***	***0.028***
	Cultivar ^b^	1	0.655	n.a.	0.285	1.000	0.715	0.180	0.500	0.753	0.753
oats	Year ^a^	2	n.a.	0.202	0.607	0.135	0.223	n.a.	n.a.	0.135	0.135
	Location ^b^	1	n.a.	0.285	***0.043***	0.180	0.068	n.a.	n.a.	0.068	0.180
	Cultivar ^b^	1	n.a.	1.000	0.893	n.a.	0.655	n.a.	n.a.	0.180	n.a.

DON–deoxynivalenol, T2–T2-toxin, HT2–HT2-toxin, NIV–nivalenol, BEA–beauvericin, ENA–enniatin A, ENA1–enniatin A_1_, ENB–enniatin B, ENB1–enniatin B_1_; ^a^ Friedman test, ^b^ Wilcoxon signed-rank test; Significant values (*p* < 0.05) are in bold and italics; n.a.–not available (particular mycotoxin was below LOQ).

**Table 3 foods-09-01037-t003:** Comparison of mycotoxin occurrence in hulless barley and oats grown at 2 experimental sites over 3 harvest years. Differences are expressed as *p* values (Mann-Whitney U Test).

			Mean Concentrations ± Standard Deviation and *p* Values								
Crop	Location	*n*	DON	T2	HT2	NIV	BEA	ENA	ENA1	ENB	ENB1
barley		12	31 ± 13	3 ± 1	10 ± 9	239 ± 218	57 ± 118	5 ± 4	32 ± 32	226 ± 69	105 ± 105
oats		12	25 ± 0	4 ± 3	11 ± 9	68 ± 77	10 ± 12	3 ± 0	3 ± 0	10 ± 5	4 ± 4
			0.166	0.113	0.832	***0.022***	0.069	0.079	***0.000***	***0.000***	***0.000***
barley	KM	6	36 ± 18	3 ± 1	18 ± 6	61 ± 51	4 ± 2	3 ± 0	11 ± 9	39 ± 31	30 ± 25
oats		6	25 ± 0	5 ± 3	16 ± 9	40 ± 0	3 ± 0	3 ± 0	3 ± 0	3 ± 0	3 ± 0
			0.378	0.262	0.810	0.689	0.378	0.936	0.066	***0.005***	***0.020***
barley	ZB	6	25 ± 0	3 ± 0	3 ± 0	417 ± 161	110 ± 154	7 ± 5	54 ± 33	413 ± 201	180 ± 101
oats		6	25 ± 0	3 ± 2	6 ± 6	96 ± 106	17 ± 14	3 ± 0	3 ± 0	18 ± 21	6 ± 5
			1.000	0.689	0.378	***0.008***	***0.013***	0.173	***0.020***	***0.005***	***0.008***

DON–deoxynivalenol, T2–T2-toxin, HT2–HT2-toxin, NIV–nivalenol, BEA–beauvericin, ENA–enniatin A, ENA1–enniatin A_1_, ENB–enniatin B, ENB1–enniatin B_1_. Significant values (*p* < 0.05) are in bold and italics; n.a.–not available.

**Table 4 foods-09-01037-t004:** Spearman’s correlations between individual mycotoxins in hulless barley and oats grown at two locations. Significant *r* values (*p* < 0.05) are in bold and italics.

	DON	T2	HT2	*NIV*	*BEA*	*ENA*	*ENA1*	*ENB*
**T2**	−0.153							
**HT2**	0.237	0.219						
**NIV**	0.009	−0.380	***−0.517***					
**BEA**	−0.122	−0.290	***−0.495***	***0.833***				
**ENA**	−0.114	−0.192	−0.372	***0.572***	***0.485***			
**ENA1**	0.227	−0.380	−0.177	***0.611***	***0.581***	***0.659***		
**ENB**	0.115	−0.333	−0.321	***0.592***	***0.718***	***0.530***	***0.826***	
**ENB1**	0.208	−0.286	−0.206	***0.603***	***0.695***	***0.604***	***0.914***	***0.948***

DON–deoxynivalenol, T2–T2-toxin, HT2–HT2-toxin, NIV–nivalenol, BEA–beauvericin, ENA–enniatin A, ENA1–enniatin A_1_, ENB–enniatin B, ENB1–enniatin B_1._

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
