# Peer review of "Fusarium Mycotoxins in Two Hulless Oat and Barley Cultivars Used for Food Purposes"

_foods, 2020, doi:10.3390/foods9081037_

Round 1

Reviewer 1 Report

It is a research work carried out by an expert in the subject in question.

The text is neat and well written, following the rules of the Journal.

The summary is correct and complete, since it perfectly fulfills its function. The introduction is interesting and allows us to understand the existing problem and what is the objective of the work. The Material and Methods I especially liked for explaining in detail the methodologies used. The presentation of the results, text and figures, is correct. The discussion is also complete but difficult to read. And the conclusions confirm the interest of the results and their relevance due to the novelty of the study.

Here are some comments to improve the text:

Lines 106-111: This is information that must be in Material and Methods.

Lines 169-173: Climate data should appear in Material and Methods and then should be commented in the Discussion. But they should not appear in the Results.

Lines 243-391: The Discussion is difficult to read. To understand it better, I consider that subsections should be made, similar or equal to those of the Results, and include in each one the corresponding discussion.

Author Response

Reviewer 1:

Q: Lines 106-111: This is information that must be in Material and Methods.

A: This part of the text was reformulated to be more concise and corresponding with the position in the text. The current text is:

„ The aim of our study was to investigate mycotoxin contamination of hulless barley and hulless oats grown under different environments and to compare the concentrations with the legal limits for food cereals (if existing).“

The remaining information has been included into Materials and Methods, as you suggested.

Q: Lines 169-173: Climate data should appear in Material and Methods and then should be commented in the Discussion. But they should not appear in the Results.

A: The paragraph containing climate data was transferred into Materials and Methods and included in paragraph describing experimental locations. Relationship with the results (mycotoxin content) is commented in the Discussion (Paragraph 4.6 Factors Influencing Mycotoxin Content of Hulless Barley and Oats, Lines 494-603 (of the revised text)

Q: Lines 243-391: The Discussion is difficult to read. To understand it better, I consider that subsections should be made, similar or equal to those of the Results, and include in each one the corresponding discussion.

A: The Discussion was restructured in order to be more readable and to present the results in a logical way. Subsections were made according to results for individual mycotoxins (or their groups), and also one subsection related to the main differences between hulless barley and oats and one section related to factors influencing mycotoxin content of hulless barley and oats. Some duplications between text in Discussion and Results have been eliminated. The numbering of the References were adjusted accordingly.

Thank you for all remarks and for encouraging evaluation. I really appreciate your review.

Ivana Polišenská and co-authors

Reviewer 2 Report

The topic of MS is interesting and undoubtedly important. The authors have applied proper analytical methods and I have no major comments in this respect. However, I have some general remarks concerning the experiment and statistical methods used.

  1. The studies on natural contamination of cereal grain by mycotoxins are always burdened with some “environmental” errors (variability of weather conditions, inoculum potential resulting from the natural presence of specific Fusarium species, etc.). Therefore, it is very good that the authors present the results of three-year experiment. However, in such an experiment it would be worthwhile to include the reference variety or varieties (not so important whether naked or hulled but with known resistance to the accumulation of toxins in the grain).
  2. What do the authors mean by "The homogenous sample" (line 140)? In how many replications were chromatographic analyses made? Were separate chromatographic analyses made of all the field replicates (biological replications) or were applied standardised bulk samples from three replicates (technical replications)? Was the concentration of moniliformin, a metabolite characteristic mainly (but not only) of F. avenaceum, determined? This would be very interesting because of identification of hexadepsipeptides in the grain samples. Both F. avenaceum and F. poae but also F. tricinctum have been reported to produce ENNs and BEA.
  3. The statistical analysis of the results of mycotoxin concentration is always difficult, mainly due to the lack of the normality of distribution of data. The authors state that the Shapiro-Wilk test was used to check the distribution normality as well as data were log-transformed. My questions: (1) did this test determine the normal distribution for each metabolite; (2) is the log-transformation a sufficient tool to normalize the data in case of the results obtained and presented by the authors? The normality of distribution is a sine qua non condition to ANOVA.
  4. Table 4. In my opinion, the application of a simple correlation for such very specific results (for barley 50 samples < LOQ and for oats 92 samples <LOQ, which is 46 and 85% respectively) is incorrect. For such results, as better alternatives to Pearson correlation and ANOVA would be to use one of the non-parametric methods. Moreover, it would be worth to use multidimensional analysis (e.g. PCA or cluster analysis) for standardized or ranked data.

Specific comments:

  • Figure 1 is poorly legible. Both the font and designations/symbols are too small.
  • Table 2 - If the authors want to use colorful fill in the table cells, then "dark green" is unnecessary. Only values p<0.01 and <0.01p<0.05 could be highlighted.
  • Table 3 – The only probability values alone do not give sufficient information here. The information that the means differ significantly is not enough because the reader does not even know which means were higher and which were lower.
  • Table 4 – see comment #4 above

Author Response

Rewiever 2

  1. The studies on natural contamination of cereal grain by mycotoxins are always burdened with some “environmental” errors (variability of weather conditions, inoculum potential resulting from the natural presence of specific Fusarium species, etc.). Therefore, it is very good that the authors present the results of three-year experiment. However, in such an experiment it would be worthwhile to include the reference variety or varieties (not so important whether naked or hulled but with known resistance to the accumulation of toxins in the grain).

A: Looking at the results from today´s point of view I have to admit that including such a standard variety would help to elucidate the level of resistence/susceptibility of currently employed varieties (AF Lucius and AF Cesar). But the main aim of the trial was to conduct research in terms of mycotoxin burden for the consumer, that means, we took the legislative limits as an absolute benchmark and intended to focus primarily on legally limited mycotoxins. To improve the information ability of the trial, apart from 3 harvest years, it was carried out at 2 experimental sites with contrasting environment. The reason is the dominance of environmental factors among mycotoxin influencing factors.

  1. What do the authors mean by "The homogenous sample" (line 140)? In how many replications were chromatographic analyses made? Were separate chromatographic analyses made of all the field replicates (biological replications) or were applied standardised bulk samples from three replicates (technical replications)? Was the concentration of moniliformin, a metabolite characteristic mainly (but not only) of F. avenaceum, determined? This would be very interesting because of identification of hexadepsipeptides in the grain samples. Both F. avenaceum and F. poae but also F. tricinctum have been reported to produce ENNs and BEA.

A: Separate chromatographic analyses were made all the field replicates. In the text it is mentioned e.g. in the part 2.3 Data Analyses (Line 179 of the revised text: „Analytical data are reported as the mean ± standard deviation of four field replications.“). See also our primary data - Table S1, S2). Moniliformin was not analysed, unfortunately. The information about moniliformin and its place among the emerging mycotoxins is included in the Introduction (Line 78), production of moniliformin by F.avenaceum is mentioned in the Discussion (Line 589). The information about production of enniatins and beauvericine by F. tricinctum was added to appropriate place (Discussion, Line 590).

  1. The statistical analysis of the results of mycotoxin concentration is always difficult, mainly due to the lack of the normality of distribution of data. The authors state that the Shapiro-Wilk test was used to check the distribution normality as well as data were log-transformed. My questions: (1) did this test determine the normal distribution for each metabolite; (2) is the log-transformation a sufficient tool to normalize the data in case of the results obtained and presented by the authors? The normality of distribution is a sine qua non condition to ANOVA.

The true is that not all metabolites were normally distributed. Log-transformation seemed to be a common way in the analysis of mycotoxin data. Although in our case this method did not always ensure strict normality, the results were in quite a good agreement with non-parametric testing. But we recognize that non-parametric statistics are more appropriate in our case. This fact is now reflected in the article, the data analyses have been performed anew with appropriate non-parametric methods (Friedman test for years, the Wilcoxon signed-rank test for localities and cultivars, the Mann-Whitney U Test for crops). The related parts of the text have been changed (Materials and Methods – part Data Analyses; Results – Tables 2,3; Discussion – the interpretation of the results).

  1. Table 4. In my opinion, the application of a simple correlation for such very specific results (for barley 50 samples < LOQ and for oats 92 samples <LOQ, which is 46 and 85% respectively) is incorrect. For such results, as better alternatives to Pearson correlation and ANOVA would be to use one of the non-parametric methods. Moreover, it would be worth to use multidimensional analysis (e.g. PCA or cluster analysis) for standardized or ranked data.

Non-parametric Spearman´s correlations have been used now instead of Pearson´s. Table 4 has been changed appropriately.

Q: Figure 1 is poorly legible. Both the font and designations/symbols are too small

A: Both font and symbols were changed from the size 12 to size 20. It really looks much better.

Q: Table 2 - If the authors want to use colorful fill in the table cells, then "dark green" is unnecessary. Only values p<0.01 and <0.01p<0.05 could be highlighted.

A: Colours are not necessary. The information ability of p values is sufficient. Significant values (p <0.05) are now in bold and italics. It is more clear.

Q: Table 3 – The only probability values alone do not give sufficient information here. The information that the means differ significantly is not enough because the reader does not even know which means were higher and which were lower.

A: Table 3 has been modified – apart from numeric p-values, also means of mycotoxin values have been incorporated. Only p values below 0.05 are now highlighted (bold and italics).

Q: Table 4 – see comment #4 above

A: Table 4 has been changed appropriately. Non-parametric Spearman´s correlations have been used now instead of Pearson´s.

Thank you for your valuable input for the paper and for the time you spent on it. We appreciate your approach very much and we are convinced that your input improved the quality of the paper substantially.

Ivana Polišenská and co-authors

Round 2

Reviewer 2 Report

I don't have any further comments on this version of the manuscript. I would like to thank the authors for responding to my previous comments.

Author Response

Dar Reviewer,

thank you for the favorable evaluation. Thank you also for the overall cooperation and your valuable input for the paper.